# A survey of mapping algorithms in the long-reads era

Kristoffer Sahlin[1*], Thomas Baudeau[2], Bastien Cazaux[2] and Camille Marchet[2*]

*Correspondence:
ksahlin@math.su.se;
marchetcamille@gmail.com

[1] Department of Mathematics, Science for Life Laboratory, Stockholm University, 106 91 Stockholm, Sweden
[2] Univ. Lille, CNRS, Centrale Lille, UMR 9189 CRIStAL, F-59000 Lille, France

### Abstract

It has been over a decade since the first publication of a method dedicated entirely to mapping long-reads. The distinctive characteristics of long reads resulted in methods moving from the seed-and-extend framework used for short reads to a seed-and-chain framework due to the seed abundance in each read. The main novelties are based on alternative seed constructs or chaining formulations. Dozens of tools now exist, whose heuristics have evolved considerably. We provide an overview of the methods used in long-read mappers. Since they are driven by implementation-specific parameters, we develop an original visualization tool to understand the parameter settings (http://bcazaux.polytech-lille.net/Minimap2/).

## Introduction

With the introduction of PacBio long-read sequencing and later Oxford Nanopore Technologies, a need for mapping long and noisy sequencing reads emerged. The data proposed new computational challenges of mapping millions of sequences, initially at expected error rates of 10–20%. In addition, researchers noticed that the seed-and-extend paradigm used in short-read mapping was not practical for long-reads. First, seed-and-extend would usually rely on a single match before extending, while long-reads required multiple consistent matches along the read to be confidently mapped. Second, the extending part, which relies on pairwise alignment algorithms with quadratic time complexity, had to be avoided, given the combined length and the frequent insertions and deletions in long-read data. Early on, the computational problem was compared to whole-genome alignment, with the additional complexity of high error rates. Such observations lead to the novel *seed-and-chain* paradigm for mapping long-reads (see Fig. 1). However, the first long-read mapping algorithms using older seeding techniques designed for generic sequence alignment (e.g., BLAST) were not time-competitive in their throughput compared to short-read mappers. Thus, sketching techniques imported from comparative genomics started to appear in this domain.

**Fig. 1** Differences in the main steps between short-read mapping (left) and long-read mapping (right). *Query* denotes the read and *reference* denotes a genome region. Mainly, short-read approaches extend (orange parts) from a single anchor (in blue) on the whole read length while long-read approaches gather multiple anchors, and chain (yellow line) them in for a candidate extending procedure that is done between pairs of anchors

Recently, specific sub-problems in the mapping domain have been identified and investigated, such as partial and gapped extension alignment of reads for structural variant discovery, mapping reads in repetitive regions or from non-reference alleles to correct loci, and other applications such as spliced-mapping of RNA reads. These specific problems require and motivate novel algorithmic solutions. In this survey article, we give an overview of the techniques proposed over the last decade for mapping long reads to genomes. After giving definitions and main intuitions, we describe the methodology in two steps. We first discuss seeding, up to the latest advents using novel seeds (e.g., syncmers and strobemers). We then discuss chaining, for which we decipher the currently used score functions. We also made available an original visualization tool that can be used to play with the different parameters in order to understand their impact on the chain (http://bcazaux.polytech-lille.net/Minimap2).

## Definitions and state-of-the-art of tools

### Preliminaries

In this survey, we restrain ourselves to the problem of mapping a sequence shorter or equal to a genome (a read) to a reference genome. We further assume that reads come from a genome that is closely related to the reference genome, such as from the same organism or a closely related species.

Let $q = (q_1, \ldots q_l)$ be the read sequence of size $l$ and $t = (t_1, \ldots t_n)$ the sequence of the reference region of size $n$. Let $\Sigma = \{A, C, G, T\}$ and $\Sigma_+ = \{A, C, G, T, -\}$ be two alphabets, $x$ and $y$ strings are defined on $\Sigma$. Let $f : \Sigma_+^* \to \Sigma^*$ be a transform that maps a string to its subsequence with all "$-$" characters removed. An alignment is a pair of strings $(q', t')$ such that:

1. $q'$ and $t'$ have the same size: $|q'| = |t'| = S$
2. The initial sequences are retrieved through the transform: $f(q') = q$ and $f(t') = t$
3. Any pair of characters can be matched at a position $i$ of the strings but two dashes: $(q'[i], t'[i]) \neq (-, -)$, for $0 \leq i < S$

Many alignments exist for a given pair of strings, in theory, the methods described hereafter aim at finding *good* alignments, i.e., alignments that optimize some distance between the pair of strings. The distance is computed using score functions which give rules on the characters pairing. Algorithms exist to compute optimal semi-global pairwise alignments between a read and the reference genome with respect to a score function. However, their complexity is $\mathcal{O}(n \times l)$ and disqualifies them in the context of handling big data such as sequencing data. Therefore, methods in the literature use heuristics to narrow down a set of candidate locations before performing pairwise alignment. This heuristic procedure has been commonly referred to in literature both as *read mapping* and *read alignment*.

We will in this survey refer to *read mapping* as the complete procedure of finding the read's location on the genome (through seeding and chaining steps) and extending the alignment between the read and genome region identified by the mapping location by pairwise alignment. The mapping algorithms we discuss do not guarantee to find the optimal solution. In case we discuss the procedure of only finding a reference location for the read without the alignment extension, we refer to the procedure as *extension-free mapping*.

In our survey, we discuss read mapping to a genome sequence. We will use the terms *query* for a read and *reference* to denote the genome.

### Overview of fundamental ideas

To our knowledge, the first mappers explicitly written for long-reads were YAHA [29] and BLASR [15], although short-reads mappers had been adapted for the long-read usage [53, 57, 64]. While solutions specialized for either Nanopore [5] or PacBio [38] characteristics appeared, most modern mappers work for both technologies with adapted parameters. BLASR presented itself as a hybrid approach descending from both genome-to-genome alignment methods (such as MUMmer [20]) and short-read mappers. The paper contains seminal ideas used in modern long-read mappers such as the seed-and-chain paradigm.

#### *Seeding*

Seeding is the first operation in the heuristics used by mapping techniques.

**Definition 1**    A **seed** is a subsequence extracted from the query or the reference.

The purpose of seeding is to find relatively small matching segments between the query and the reference that serves as markers for reference regions that potentially are similar to the read. The reason seeding is used is that it is typically computationally efficient to find matching seeds that can narrow down regions of interest compared to, *e.g.*, global pairwise alignment of the read to the reference. As we will see in the "Seeding almost always uses sketched, exact, fixed-length seeds" section, seeds can be of different nature. Seeding relates to pattern matching, although in sequence bioinformatics, practically all approaches work under the paradigm which indexes the reference and query the index to find matches. The underlying assumption is that once the index is created, it can be used several times to map different query sets. To save space, reference indexes

can be in a compressed form. Once matches are found, a second operation aims at finding sets of concordantly ordered seeds between the query and the reference (*chaining*; "Chaining is dominated by dynamic programming with concave gap score functions" section) and to "fill the gaps" between seeds as well as providing the final nucleotide level alignment (*extension*; "Extension step and final alignment computation" section). Seeding was quickly identified as a critical phase in long-read mapping, which led to novel proposals [58, 66, 95].

### Sketching

An important idea for seeding is *sketching* that was introduced in MHAP, a long-read overlap finder implemented in an assembly algorithm [7]. The rationale was to improve the time efficiency of the long-read mapping problem in comparison to the throughput of the second generation sequencing mappers. Sketching consists of compressing the information of a set (here a set of *k*-mers) into a fixed-length vector (a sketch) of representative elements called fingerprints. By comparing two sketches, one can approximate a similarity estimation of the two sets quickly and independently of their initial set sizes. Several approaches exist [10, 17, 73]. MinHash [10] is a sketching technique based on locally sensitive hashing, which produces an unbiased estimator for the Jaccard distance between two sets by comparing a subset of items in a very efficient way. MHAP relied on sketching with this MinHash approach. Thus, MHAP overcame a space limitation of BLASR which would index the whole reference. The type of matches (exact, fixed-size) induced by MHAP's approach also allowed to perform rapid queries. An important limitation of MHAP was that the sampling technique gave no guarantee to uniformly cover the query's sequence. In other words, there was no guarantee on the maximum distance between two consecutively sampled seeds. This led to the development read mappers that used sketching techniques with guarantees on maximum distance between sampled seeds, starting with minimap [58]. Sketching is still an active research area of long-read mapping with several recent developments [22, 31, 48, 91].

### Chaining

A key intuition is that in short-reads mapping, the extending procedure could start after finding a single shared seed between the query and the reference, called anchors (for details on techniques related to the previous sequencing generation, we refer the reader to a methodological survey of short-read mapping [3]).

**Definition 2**   An **anchor** is a matching seed between the query and the reference. It is represented by a pair of coordinates on the query and the reference.

In the literature, an anchor can also be called "a fragment" or "a match." Two anchors are said to overlap if one anchor starts or ends within the coordinate interval defined by the other anchor. In long-read mapping, the length of the reads and the short seed length used due to the initial high long-read error rates can lead to a large number of anchors. It is therefore necessary to reduce the search space by selecting subsequences of ordered anchors (chains).

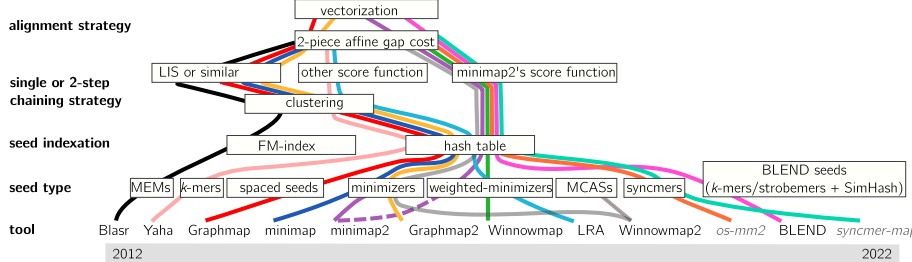

**Fig. 2** Long read mapping tools over time. Tools and techniques are presented from oldest to most recent, from left to right. The figure presents implementation names at the bottom, then goes up to the different steps: seeding, with seed selection strategies and indexation, then chaining and pairwise alignment strategies. The dotted line for `minimap2` means its implementation evolved from strategy in plain line to strategy in dotted line. The gray italic names denote for proofs-of-concept rather than tools

**Definition 3**    Let $\mathcal{A} = [a_0, a_1, \ldots, a_k]$ be a list of anchors defined by their coordinates on the reference and the query. A **chain** is a subset of $\mathcal{A}$ of length $c \leq k$. A colinear chain is a subset of $\mathcal{A}$ in which anchors are sorted by such that if $i < j$, $a_j$ an anchor of starting coordinate $(x_j, y_j)$, with $x_j > x_i$ and $y_j > y_i$, $(x_i, y_i$ the ending coordinate of $a_i)$ in the (*reference, query*) plane.

Drawing inspiration from genome-wide mapping, `BLASR` introduced a chaining step which aims at selecting high-scoring chains from a set of candidate chains. Chaining allows to reduce the final step of a long-read mapper (the base level extension) to pairwise alignment of sub-regions between ordered anchors in chains. Chaining in long-reads has been solved using various dynamic programming procedures [59, 82, 95]. In particular, the continuous work effort provided in `minimap2` [58–60] in both seeding and chaining processes made it a baseline for many other tools' development. Figure 2 shows the different algorithmic choices over time for seeding and chaining.

While this survey covers the genomic mapping aspects, other important contributions have dealt with adapted procedures in the case of long-read RNA mapping [67, 72, 86, 99], and structural variant identification [33, 65, 89, 98], or alignment through large repeats [12, 74]. Other related research focused on read-to-read overlap detection [26, 100][1], or extension-free (pseudo-alignment) approaches [16, 25, 46]. Finally, here we describe algorithmic solutions working on the nucleotide sequence, but raw signal mappers for Nanopore long-reads is also an active area of research [39, 54, 101].

In the following, we hardly elaborate on complexities for the different algorithms. Some are yet unknown, but in many cases implementations simply use heuristics so that each step's time is expected to be linear.

## A survey of algorithmic steps

### Seeding almost always uses sketched, exact, fixed-length seeds

Seeding is the procedure that consists in collecting a set $\mathcal{S}$ of seeds from the reference, then finding anchors between the query's seeds and $\mathcal{S}$. In order to find anchors

---

[1] and the unpublished `DALIGNER` https://github.com/thegenemyers/DALIGNER

efficiently, $\mathcal{S}$ is stored using an index data-structure. In the following sections we detail the different types of seeds that have been used in long-read mapping.

### k-mers

Substrings of length *k*, or *k*-mers, are perhaps the most commonly used seed in bioinformatics. A *k*-mer seed can be indexed by using a hash function to produce an integer value (usually as a 32 or 64-bit integer), which is then added to a hash table. This makes indexing of *k*-mers computationally cheap, provided that the hash function and hash table implementations are efficient. Methods to efficiently hash *k*-mers have been proposed [75], which uses the previous *k*-mers hash value to compute the next one using a rolling hash function.

If a *k*-mer anchor is found, it is guaranteed to be exact (disregarding hash collisions). While it is desirable to produce anchors only in identical regions to minimize hits to non-homologous regions, a downside is that mutations in homologous regions will also alter the *k*-mers, preventing anchors in the region. Typically, a single substitution alters $2k - 1k$-mers. The length distribution of stretches of consecutive non-matching *k*-mers between two homologous regions with substitutions depends on the substitution rate, and has been studied theoretically in [8].

### k-mer sketching

As two consecutive k-mers share most of their sequence, they are mostly redundant. Therefore, we could reduce the memory overhead and query time without losing much information if only some k-mers were stored. Here we present different methods for picking a subsample of representative *k*-mers as seeds. These approaches have proven their efficiency by drastically reducing the number of objects to index while keeping high sensitivity and specificity in mapping applications. There exist two broader classes of sketching techniques, methods that offers a distance guarantee between consecutively sampled seeds, and methods that does not. Both classes of methods have been used to estimate similarity between sequences. However, the central research questions of the former class of methods involve the distance distribution between sampled seeds, the fraction of conserved seeds under mutations, and the compression ratio to original input data. In contrast, the central studied question for sketching methods without distance guarantees is often to produce unbiased estimations of sequence identity [10, 40, 69]), which distance bounded sketching methods can not guarantee [6].

*No guarantee on distance*   In this category methods typically stores a predetermined number of *k*-mers (e.g., MinHash [10], OrderMinHash [69]) from the sequence. The *k*-mers are selected based on the property of their hash value. For example, in Min-Hash sketching, a total ordering on the *k*-mers' hashes is used, and a fixed set of minimal hashes in the ordering are kept. This technique gives no distance guarantee between seeds, meaning a large gap can appear between two consecutive sampled *k*-mers. MinHash has been used to perform extension-free mapping [46] for genome-length sequences and to find read-to-read overlaps in long-read assembly [7, 90]. However, fixed-size sketches do not adapt well to different read lengths. The number of sampled seeds remains constant for any number of distinct *k*-mers. Because of this, two similar

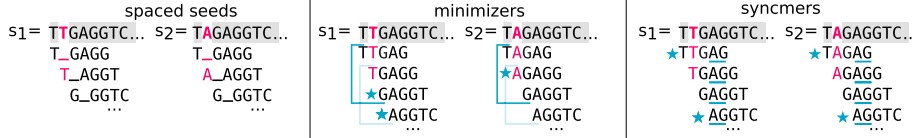

**Fig. 3** Illustration of spaced-seeds, minimizer selection, syncmer selection and context dependency. Here, two sequences $s_1$ and $s_2$ are different from a single mutated base (second base, in pink). When comparing those sequences, one would like to focus on common bases, i.e., bases highlighted in grey. In the left panel, we present spaced-seeds for $k = 5$, with a wildcard at second position (represented by an underscore). We observe that identical spaced-seeds can be spelled over a mutated locus, e.g., `T_GAGG`. In the middle panel, we present a selected minimizer with $k = 5, w = 3$. One blue window is presented, a second is suggested in lighter blue. A star shows the position of the selected $k$-mer in the window (we use lexicographic order). The mutated base has an impact on the overall window content, therefore a $k$-mer from the (unmutated) region of interest in $s_1$ is no longer selected in $s_2$. On the contrary, in the right panel, we show that syncmers can be more robust in this situation. We choose $k = 5, s = 2$ and present closed syncmers. We underline the smallest $s$−mer in each $k$-mer in blue and a star shows the selected $k$-mers. We see that in this example, the mutated base has no impact on the syncmer selection, and the same syncmer is selected in the region of interest for $s_1$ and $s_2$

regions from sequences of different sizes will not automatically have the same selected seeds, which is a desired property for seeding. Therefore this approach was later replaced by other scaled sketch strategies [40, 45]. FracMinHash has been used for long read mapping [25] (called universal minimizers in their study), and works well when reads are long enough, but it is important to note that theoretically there does not exist a distance guarantee for scaled sketch hashing methods, regardless of the density of the sketch.

*Distance guaranteed*   The first distance bounded $k$-mer sketching technique proposed for long-read mapping was *minimizers*. Minimizers have been introduced in two independent publications [83, 88], and was popularized in bioinformatics by the tools `minimap` [58] and `minimap2` [59]. In our framework, minimizers are $k$-mers sampled determined by three parameters $m$, $w$, and $h$. $h$ is a function that defines an order, e.g., the lexicographical order. Given the set of $w$ consecutive $k$-mers in a window at positions $[m, m + w − 1]$ on the sequence, a minimizer is the $k$-mer associated with the minimal value for $h$ over this set (see left panel in Fig. 3). Minimizers are produced by extracting a minimizer in each consecutive window $w \in [0, |S| − w + 1]$ over a sequence $S$.

Since at least one minimizer is selected in each window, they have a *distance guarantee*. While the distance guarantee (hence seed density in regions) is desired for mapping applications, it is also desired to sample as few minimizers as possible to reduce storage. Different optimizations have been proposed to reduce the density of sampled minimizers while keeping the distance guarantee. Weighted minimizers [48] implement a procedure to select $k$-mers of variable rareness. In order for $k$-mers from highly repetitive regions not to be as likely as others to be selected, it first counts $k$-mers, and down weights the frequently occurring ones. Then it takes this weight into account for the hashing procedure. If low occurrence $k$-mers are too far away in a query, a solution [60] allows sampling minimizers also in the repetitive region by keeping some of the lowest possible occurrences among the minimizers in the repetitive region.

In minimizer sketching, the choice of the minimizer in each window depends on the other *k*-mers in the window. This property is called *context dependency* [91]. Context dependency is typically not desired in sketching methods as neighbouring *k*-mers affected by mutations may alter the minimizers in a window. However, for finding anchors it is desired to guarantee that the same *k*-mers are sampled between two homologous regions regardless of neighboring mutations. Therefore, *context-independent* methods have been proposed such as syncmers [23] and minimally overlapping words (MOW) [32], where the sampling does not depend on the neighboring *k*-mers. Syncmers was used in the context of long-read mapping [91] in an alternative implementation of `minimap2` and even more recently in [22][2]. For their construction, syncmers use *s*-mers of size $k - s + 1$ $(s < k)$ occurring within *k*-mers (see right panel in Fig. 3 for an illustrated difference with the minimizers, and Additional file 1: Fig. S1). The *k*-mer is selected if its smallest *s*-mer meets some criteria. An example criteria is that the *s*-mer appears at position *p* within the *k*-mer $(0 \leq p < k - s + 1)$ (these are called *open syncmers*), a more studied category is *closed syncmers* where *p* must be the first or the last *s*-mer position in the *k*-mer. This way of selection uses properties intrinsic to each *k*-mer, therefore is context-free. Closed syncmers also have a distance guarantee. By construction, syncmers tend to produce a more even spacing between sampled seeds while still allowing a distance guarantee.

### Fuzzy seeds

Due to read errors and SNPs between the reference and sequenced organism, it is in many scenarios desired that a seed anchors the query and the reference in homologous regions even if the seeds extracted in regions differ. In other words, we would want similar seeds to hash to identical hash values. A hash function that produces identical hash values for similar but not necessarily identical inputs is usually called a locality-sensitive hash function. We will refer to seeds produced under such methods as fuzzy or inexact seeds. Several methods to produce fuzzy seeds have been described.

Perhaps the most common one is spaced seeds. Within a spaced seed, some positions are required to match (called fixed positions), while the remaining positions can be ignored (called wildcards or don't care positions). Within a *k*-mer, fixed positions can be selected as wildcards by applying particular masks on the *k*-mer's bases [44]. Spaced seeds are effective for data with substitutions and are, for example, used in the popular sequence mapping software BLAST [4], metagenome short-read classification [13], and in long read mapping tool GraphMap [95]. Typically, multiple seed patterns are used [62, 95] where the overlap between the fixed positions in the seeds should be minimized [43] to increase sensitivity. For example, GraphMap queries three different seeds to the index for each position in the query. This design is capable of handling substitutions and indels of one nucleotide. We provide details on this scheme in Additional file 1: Figs. S2 and S3. However, spaced seeds can only handle indels if multiple patterns are queried per position, and the number of patterns required increases with indel size [35]. Although the computation of *good* sets of spaced seed patterns has been optimized [44], using such

---

[2] https://github.com/bluenote-1577/os-minimap2 and https://github.com/Shamir-Lab/syncmer_mapping

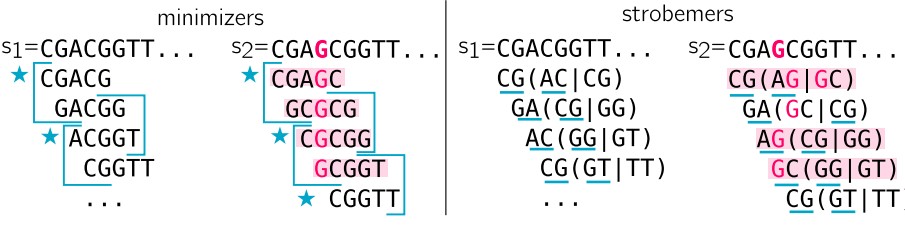

**Fig. 4** Illustration of strobemers' capacity to handle indels. As in Fig. 3, two sequences are presented. This time, $s_2$ has an insertion (pink G). On the left panel, minimizers are selected using $w = 2, k = 5$. Blue stars point selected minimizers in each blue window. One can see that the only safe region to generate minimizer is the CGGTT sequence after the insertion, that is shared and of length $\geq k$. Put differently, $k$-mers in red have no chance to be in common between the two sequences. However, in this example, the scheme fails to select a common minimizer in the safe region. Strobemer selection is presented in the right panel, using $k = 2, s = 2, w = 2$. At each position, the first $k$-mer is selected to be the start site of the strobemer. Then, in the non-overlapping window (of size $w$) downstream to the first $k$-mer, a second $k$-mer is selected according to one of the selection techniques presented in [84] (we illustrate selecting the lexicographical minimizer). We underline the bases that are kept for each strobemer. For instance in $s_1$, the first $k$-mer is CG at positions 0 and 1, then the next window starts at position 2. Two $k$-mers are computed from this window, AC and CG, and AC is the minimizer. Therefore, the strobemer is (CG,AC). Again, strobemers with no chance to be shared between $s_1$ and $s_2$ are colored in red. For strobemers, it is the case when at least one part contains the mutated base. We note that not only the CGGTT region has a common strobemer (CG,GT) in both sequences, but also that the scheme allowed to "jump over" the mutated G and could select another common strobemer (GA,CG) in a more difficult region. The strobemers in this example consists of two $k$-mers ($s = 2$) but they can be constructed for other $s > 2$

seeding can become computationally prohibitive if the application requires to match over indels beyond a couple of nucleotides.

As indels are a frequent source of variability on long-reads, spaced seeds have, except GraphMap, not been frequently used in long-read mapping algorithm designs. There are other types of fuzzy seed constructs, such as permutation-based seeds [56], but they only tolerate substitutions and have been used in short-read mapping.

Traditionally, anchoring over indels has typically been solved by querying multiple seeds in a region and performing post-clustering of nearby anchoring seeds, which are then inferred as an anchoring region. Such an approach usually provides gold standard sequence similarity queries [4, 52]. However, it comes at a substantial computational cost, not only because of the post-clustering step but in addition because relatively short seeds must be used to guarantee sensitivity, which can yield many anchors to other regions.

To remove the overhead of post-processing of nearby seeds, one can instead *link* several $k$-mers into a seed and represent it as a single hash value before storing it in the index. Such linking techniques has recently become popular in the long-reads era, where indels are frequent. One proposed method is to link two nearby minimizers [18] or several MinHash seeds [25] into a seed. Linking several minimizers into a seed is usually a relatively cheap computation as the minimizers constitute a subset of the positions on the reference. Such seeding techniques have been used in long-read mapping [25], and long-read overlap detection in genome assembly [18] and error correction [87]. A downside with these methods is that the linking of nearby minimizers or MinHash seeds implies that if some of them are destroyed due to mutations in a region, all the seeds in the region will be destroyed. Put another way, nearby seeds share redundant information (in the form of shared minimizers or MinHash seeds). Therefore, alternative approaches

such as strobemers [84] (see right panel in Fig. 4) have been described, where the goal has been to reduce the information between close-by seeds by linking *k*-mers at seemingly random positions within a window. Such pseudo-random linking of *k*-mers implies that, if one seed is destroyed due to a mutation, a nearby seed may still anchor the region. Strobemers have been shown effective at finding anchors between long-reads and for long-read mapping [84], and have been used in short-read mapping programs [85], but the pseudo-random linking come at an increased computational cost to simple linking of neighboring *k*-mers.

Two other indel tolerant fuzzy seeding techniques are `BLEND` seeds [31] and Tensor-Sketch [50]. The `BLEND` seeding mechanism mixes SimHash [17] (an alternative locality sensitive hashing to MinHash) applied either to minimizers or strobemers to construct fuzzy subsampled seeds. The authors showed that read mapping and overlap detection with `BLEND` seeds implemented in `minimap2` [59] could improve mapping speed and, in some cases, accuracy. TensorSketch [50] is based on computing all subsequences of a given length within a given window. The idea is that similar sequences will share many similar subsequences and lie close in the embedding space. TensorSketch has been used in long read mapping to graphs and offers high sensitivity but at a significant computational cost to approaches using exact seeds [49].

### Dynamic seeds

Previously discussed seeds share the characteristic that they can all be produced and inserted in a hash table and, consequently, only require a single lookup. Such techniques are typically fast and, hence, popular to use in long-read mapping algorithms. However, the downside is that if a seed differs in a region between the reference and the query (e.g., due to an error), there is no way to alternate the seeds in this region at mapping time. There are, however, other types of seed constructs that we here refer to as dynamic seeds that can be computed on the fly at the mapping step and then used as seeds downstream in the read mapping algorithm.

Maximal exact matches (MEMs) [20] are matches between a query and reference sequence that cannot be extended in any direction on the query or reference without destroying the match. These are typically produced by first identifying a *k*-mer match and then applying an extension process. MEMs are guaranteed to be an exact match between the query and the reference and are bounded below by length *k* but do not have an upper threshold for seed size. MEMs have been used in earlier long-read mapping programs (e.g., `BWA−MEM`) [15, 57] and for long-read splice mapping [86], but these seeds are more computationally expensive to compute and are typically slower than single-query seed-based algorithms.

*Minimal confidently alignable substrings (MCASs)*   If a query was sampled from a repetitive region in the reference, one might likely find several clusters of anchoring seeds across the reference. Further dynamic programming operations to decipher the true origin region of the query are typically costly or even unfeasible if too many copies have to be considered. The query might also be attributed to the wrong copy because of the sequencing errors. A recent contribution [47] proposed a solution for seeding in repetitive regions. The procedure finds the smallest substrings that *uniquely* match

(MCASs) between the query and the reference. There can be as many as the query length in theory. In practice, the more divergent the repeats, the shorter the MCASs, since a base pertaining to a single copy is more likely to be found.

### Implementation of the seeding step

#### *Seed transformations before indexing*

Originally, minimizers used a lexicographical ordering. However, in our four base alphabet, this can tend to select sequences starting with long alphabetically smaller runs such as "AAA…". Random hash functions assigning each $k$-mer a value between 0 and a maximum integer are preferred [88].

Long read technologies are known for accumulating errors in homopolymer regions, typically adding/removing a base in a stretch of a single nucleotide. Sequences can be homopolymer-compressed before finding $k$-mers. Homopolymers longer than a size $s$ are reduced to a single base, then $k$-mers are computed over the compressed sequence. For instance, for $s = 3$, $k = 4$, an original sequence ATTTTGAAAACC is compressed to ATGACC, and the final $k$-mers are ATGA, TGAC, GACC. This procedure allows finding more anchors while indexing fewer $k$-mers or minimizers. Homopolymer compression appears in long-read mapper implementations (e.g., [59]).

In regions of low complexity (e.g., ATATATA, CCCCC) the standard minimizer procedure keeps all minimal $k$-mers in windows. It is then possible for two $k$-mers to get the minimal value and to be selected, which tends to over-sample repetitive $k$-mers. A *robust winnowing* procedure is proposed in [48], which avoids the over-sampling effect by selecting fewer copies of a $k$-mer, but increases context dependency.

#### *Hash tables prevail for seed indexing*

Indexing of fixed size seeds is usually done using hash tables (although FM-indexes for $k$-mers exist [9]). In the context of sketching, invertible hash functions have been a key asset for using minimizers as $k$-mers representatives. In other words, a hash value is associated with one and only one $k$-mer, and the $k$-mer sequence can be retrieved from the hash value (using reciprocal operations). This choice allows a very fast $k$-mer/minimizer correspondence, but is memory-wise costly as it implies that the fingerprints of the hash table are not compressed (which is mitigated by the density of the sketching). Minimizers are then used to populate a hash table, which associates them to their position(s) in the reference and their strand information (usually hashed seeds are canonical $k$-mers: the smallest lexicographic sequence between the original $k$-mer and its reverse complement). There also exists learned index data structures [51] that further accelerates the querying of minimizers.

Variable-length seeds are indexed in full-text data structures (e.g., suffix arrays or FM-index [30]), which allow to find and count arbitrarily long queries in the reference. They have been used in the first versions of long-read mappers. However, variable-length seeds takes longer to query in these data structures, while hashed matches are queried in constant time. Since minimizers represent fixed-length $k$-mers, hash table solutions mainly prevail.

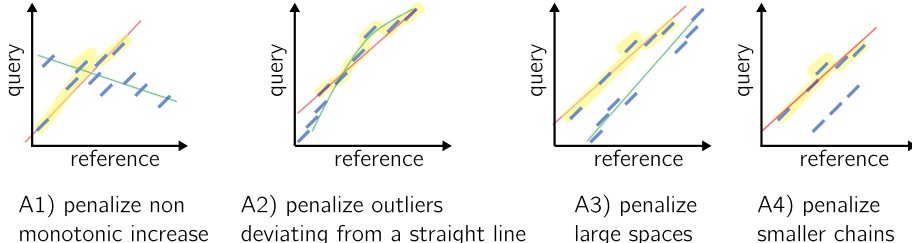

**Fig. 5** An illustration of the different constraint taken into account in the gap score functions. The reference axis shows a genome region of interest where anchors were found, not the whole reference. A1–A4 correspond to items in the text in the "A dynamic programming problem" section. Anchors are showed in blue. The selected chain with respect to the described constraint is highlighted in yellow and a line approximately passing by its anchors is showed in red. The longest chain is covered by a green line if it was not selected

### *Selecting seeds to query*

In [59], it is proposed to index all minimizers from the reference during the indexing phase (although the latest versions include weighted *k*-mers and robust winnowing heuristics), and instead skip to use highly repetitive *k*-mers to find anchors (also called soft masking). The authors noticed that in cases where a query is sampled from a repetitive region, such a procedure prevents it to be seeded. Techniques that use longer fuzzy seeds (e.g., strobemers) [31] reduce the number of masked regions, although it comes at the cost of sensitivity. Another approach [82] computes a new set of minimizers on the targeted reference regions in order to obtain finer candidate chains, which helps alignment confidence particularly in repeated or low complexity regions.

### Chaining is dominated by dynamic programming with concave gap score functions
### *A dynamic programming problem*

Once the reference's seeds are indexed, a set of seeds is extracted from the query and looked up in the index to find anchors. Anchors' positions on the query and reference are stored, as well as the forward/reverse information. Instead of directly extending the alignment between anchors, as it is done in short-read mapping, a step of chaining is added and meant to accelerate further extensions. Chaining acts as a filter and a guide for smaller extensions that need to be realized only between selected anchor pairs. Without it, too many extension procedures, most of which would be dead-ends, would have to be started.

In an ideal case, there is a unique way of ordering anchors by ascending Cartesian positions in the (*reference*, *query*) space, which passes by all the anchors. In practice, some anchors are spurious, others correspond to repeated regions and yield different possible chains. Moreover, more parameters must be taken in account. Thus, methods optimize different aspects (also illustrated in Fig. 5):

> A1) Do not allow anchors which are not ascending either by the anchors' start or end coordinates in both the query and reference (see first case in Fig. 5).
> A2) Avoid discrepancies in diagonals between anchors (second case in Fig. 5).

A3) Do not allow large spaces between consecutive anchors of the chain (see third case in Fig. 5).

A4) Favor the longest possible anchor chain (fourth case in Fig. 5).

A5) If inexact matches in seeds are possible, each seed represents a couple of intervals on the target and the query. Find a subsequence of seeds that minimize the sum of the Levenshtein distances computed on these couples of intervals (roughly, ensure that the matched regions on the target and query are as similar as possible).

The problem of finding an optimal chain using non-overlapping anchors has been called the *local chaining problem* [1], although in this application anchors can overlap. The score $f(i + 1)$ represents the cost of appending an anchor $a_{i+1}$ to a chain currently ending by anchor $a_i$. This score is often called the *gap score* in the literature, though it includes other constraints, as described above. The chaining problem for long reads seeks to find an optimal colinear chain with a positive gap score.

Mainly, methods use either a two-step approach: (1) find rough clusters of seeds as putative chains, followed by (2) find the best scored chain among the selected clusters; or work in a single pass and apply a custom dynamic programming solution to find the best anchor chain. We can start by noting that one of the first mappers dedicated to long-reads solved a global chaining problem to determine a chain of maximum score, by fixing starting and ending points (anchors) such that their interval is roughly the size of the query [15]. Such an approach would easily discard long gaps and spaces in alignments.

### Chaining in two steps

*Clusters of seeds are found through single-linkage in 2D space*    The two-step approaches rely on a first clustering step. Although it tends to be replaced by single-step chaining (see "Chaining in a single step: gap score functions" section), in the following we describe the fundamental ideas of the clustering. Methods first find rough clusters of anchors by considering a discrete (*reference*, *query*) position space. In this space, an anchor realizing a perfect match is a line of the size of the seed. This line should have a 45-degree angle, which also corresponds to the main diagonal of a (*reference*, *query*) alignment matrix. The same idea stands for a set of anchors. However, because of insertions and deletions, each small line materializing an anchor may not be on the exact same diagonal, thus realizing approximate lines in the (*reference*, *query*) space. A method from image processing has been proposed to find approximate lines in this space: the *Hough transform* [21], which makes it possible to detect imperfect straight lines in 2D space. Contrary to linear regression which would output the best line explained by the anchor distribution, here an arbitrary number of straight lines can be output and considered (see Additional file 1: Fig. S4 for an illustration). Hough transform or other similar anchor grouping algorithms ([82] proposes to delineate fine-grained clusters in order to increase the chaining specificity in repeated regions) all can be assimilated to single-linkage clustering in 2D space, which finds groups of anchors placed roughly on the same diagonal.

*Anchor chaining using longest subsequences of anchors*    The previous clustering techniques aim at finding lines in groups of anchors that can be approximately colinear. To

determine truly colinear chains, a subset of anchors can be ordered by finding a longest increasing subsequence (LIS) of anchors. Let each anchor be mapped to the order in which it appears in the reference. By crossing these map values with the order of these anchors in the query, we obtain a permutation of the set $\{1, 2, \ldots n\}$ where $n$ is the number of anchors. By using an algorithm on LIS problem, we can obtain truly colinear chains in $\mathcal{O}(n \times log(n))$.

In the case of fuzzy seeds, inexact matches are to be dealt with on top of the initial increasing chain problemin order to obtain the closest base-wise anchor chain. In this case, the problem is converted to LCSk (longest common subsequence in at least $k$-length substrings). Note that there is a correspondence between LIS and LCS. The LIS of $P$ is the LCS between $P$ and the sequence $(1, 2, \ldots c)$. In both cases, neither the longest nor the increasing requirements are sufficient to find correct anchor chains: they lack definitions for other constraints, such as distance between anchors or the possibility to allow large gaps. They are complemented with heuristics or replaced by more recent approaches in "Chaining in a single step: gap score functions" section. In addition, several methods use graphs built over anchors as backbones to the chaining and alignment steps [66, 95, 98] (one approach is described in Additional file 1). Because they would fail to take into account distances between anchors, these methods have been replaced by dynamic programming approaches relying on gap score functions.

### Chaining in a single step: gap score functions

The main drawback of the approaches previously described in "Chaining in two steps" section is that though large spaces between two anchors of a pair must be avoided, some spaces correspond to gaps in the alignment and can be kept. In order to deal concurrently with these two problems, most recent methods drop the two-step clustering and LIS to directly apply a custom dynamic programming solution. It follows the same principle as LIS, but integrates a more fine-grained gap penalty solution. It defines a cost function that grants a maximum penalty for non-monotonic increasing seed chains.

*Concave gap functions*    The cost function is designed to handle the gaps induced by frequent indels in the data. Intuitively, it is likely that indels happen in clusters of $n$ positions rather than at $n$ independent positions in the chain because some regions on the query might be particularly spurious, or because of local repeats on the reference. Therefore, the same cost is not attributed to opening a gap and extending a gap, thus a linear gap function does not fit. The choice of gap functions which are concave (verifying the *Quadrangle Inequality*) improves the time complexity by using the *wider is worse* strategy [27, 34]. In practice, these concave gap functions are affine, a set of affine functions, or a combination of affine and log functions, as proposed in [59]. We chose to present `minimap2`'s [59] gap functions in Fig. 6 as they are adopted without modifications in most current papers (with the recent exception of [82]). Chains are built by aggregating close anchors of smaller coordinates to the current anchor by penalizing the shifts compared to the main diagonal. In Fig. 6, panel a presents how the set of possible anchors to prolong the chain is selected. Panel b illustrates the dynamic function's parameters. The

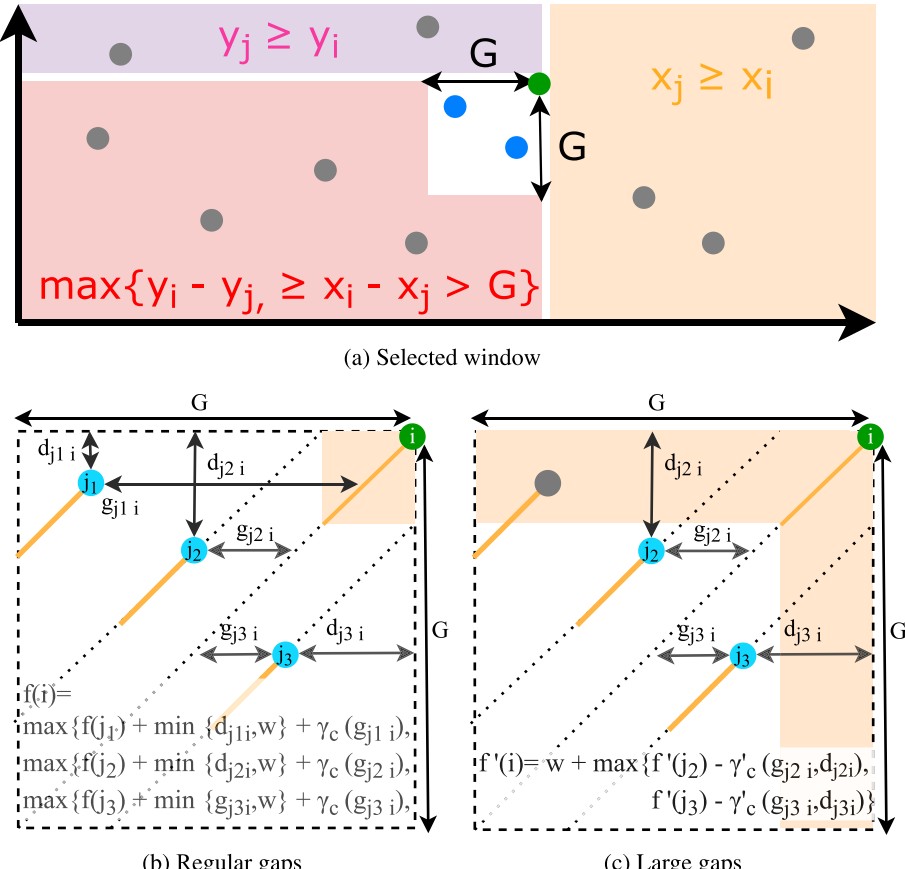

**Fig. 6** Outline of `minimap2`'s chaining. **a** shows for an anchor (in green) the selected region (in white, *G* is the gap threshold) to find available anchors to continue chaining (in blue). **b** and **c** give respectively the dynamic programming functions for regular and large gaps size. Anchors are shown as segments ending with green or blue dots with the same color code as in **a**. Besides, for the large gap size (**c**), to improve the complexity, the anchors do not overlap (available anchors are not in the red zone). $d_{ji}$ represents the smallest "distance" between the two anchors (but is not really a distance by definition), *w* is the minimizer window size, $g_{ji}$ is the gap length, and the $\gamma$ functions are the concave gap functions

complete description of the functions is available in Additional file 1. Here, we recall the formula for regular gap sizes, for anchors *i* and *j*:

$$f(i) \quad = \quad \max\{ \max_{\substack{i>j\geq 1 \\ x_i-G<x_j\leq x_i \\ y_i-G<y_j<y_i}} \{f(j) + \min\{d_{j\,i}, w\} - \gamma_r(g_{j\,i})\}, w\}$$

$$\quad\quad\quad (1) \quad\quad\quad\quad\quad\quad\quad (2) \quad\quad\quad (3) \quad\quad (4)$$

*With f(i), f(j) scores for anchors $a_i$ and $a_j$, and:*

(1) This property means that at least on axis *X* or *Y*, the distance between the two anchors must be < *G* the maximum authorized gap size, and that *i* is above and to the right of *j* (see the three colored zones in panel a of Fig. 6).

(2)This minimum penalizes overlaps between anchors. Indeed, if *j* starts before a distance of *w* on the diagonal (with *w* the minimizer window size and window guarantee of minimizers) then $d_{ji}$ (the smallest coordinate difference on either *X*- or *Y*-axis between *i* and *j*) will be lower than *w* and therefore selected. For any other case, *w* which is a larger value is selected and increases more the score.

(3)This is the concave gap score penalty (see Additional file 1 for a complete formula). It is computed on the gap length $g_{ji} = |(y_i - y_j) - (x_i - x_j)|$. It penalizes distant anchors in the Manhattan definition, i.e., anchors far from *i* either on the main diagonal or because they are on side diagonals (see for instance anchor $j_1$ in panel b of Fig. 6).

(4)This term helps with the initialization.

In order to test *w* and $\gamma_r$'s impact on the chain selection, we implemented a visualisation tool: http://bcazaux.polytech-lille.net/Minimap2/[3]. We generate a scenario of shared anchors between two sequences and allow to set the different parameters' values. We show the selected chain according to the settings.

Heuristics are applied to rapidly drop a dynamic programming procedure in regions that are unlikely to align and to avoid $O(c^2)$ worst cases. Based on empirical results, these heuristics mostly check if seeds are not separated by too large regions and drop the chaining procedure if the score becomes too low.

**Solutions for large gaps**   Noticing that [59]'s original approach would be failing in large gaps, one contribution [82] proposed techniques to perform dynamic programming with a family of concave functions by relying on a previous work [27] (built on a prior clustering step as described in "Chaining in two steps" section). Recently, [59] integrated a solution designed for mapping long structural variants in pangenomic graphs [61]. Its recent versions entail a cost function for regular gaps, and a long gap patching procedure. Then it chooses the cheapest solution to move on to the alignment step. The gap patching procedure uses a linear gap cost so that it has a higher long-gap opening cost in comparison to the regular procedure but at a cheaper extending cost. The chaining with a linear function is solved with a range minimum query (RMQ) algorithm using a balanced binary search tree [1, 81]. It allows solving the linear chaining in $\mathcal{O}(c \times log(c))$. Although, by using range maximum queue [14] instead of binary trees, this time complexity can be improved in $\mathcal{O}(c)$. As the implemented algorithm is more costly than the solution for regular gaps, the last one is preferred if possible. Panel c in Fig. 6 illustrates the dynamic function for large gaps.

---

[3] source code: https://github.com/bastcazaux/minimap2_chaining_plot_web and DOI: https://doi.org/10.5281/zenodo.7889434

*Mapping quality scores have been adapted for ranking chains*

The described methods may deliver a set of chains that satisfies the chaining score threshold. To choose among the candidates and decide the final location, chains can then be categorized into primary/secondary chains. Chains with a sufficient score are ranked from highest to lowest score. Primary chains are those with the highest scores which do not overlap with another ranked chain on the target for the most of their length. Secondary chains are others. Mapping quality, which is a measure that had been introduced to assess short-reads mapping, is redefined for long-reads with slight variations according to articles. It reports, for chains, whether the primary is very far in terms of score from the best secondary, and if it is long enough.

## Extension step and final alignment computation

While some applications, such as abundance estimation, typically need only the mapping location of a read, many applications, such as SNV and structural variation detection, rely on base-level alignments. The extension alignments step is typically computationally costly, where traditional Needleman-Wunch [78] and Smith-Waterman [94] based approaches have a time complexity $O(nm)$ if $n$ and $m$ are the lengths of the query and reference sequences, respectively. Even after locating a candidate region to extend the alignment (using the seeding and chaining steps), the quadratic complexity is still prohibitive for long reads in computation time and memory. Therefore, long read mappers typically extend the alignment of sequence segments between neighboring anchors (piece-wise extension) [59, 82, 89]. An additional benefit of piece-wise extension is that, if the chaining step supports multiple chains of disjoint regions on the query, the read mapper can output alignment to several regions, which helps detection of structural variations [59, 82, 89]. As structural variation detection is a common downstream application of long-read mapping, piece-wise extension is considered standard practice in the area [29, 59, 65, 82, 89].

Piece-wise alignment can still be computationally costly if the distance between anchors is large enough. Recently, the Wave Front Alignment (WFA) algorithm [71] made a breakthrough in both time and space complexity, where the original formulation guarantees an optimal alignment in time $O(ns)$, proportional to the read length n and the alignment score s, using $O(s^2)$ memory. Instead of a top-to-bottom traversal of the DP matrix, the WFA algorithm traverses it diagonally. It computes only the cells on the diagonals with the current highest scores (or, rather, the lowest penalties in the WFA formulation), which omits visiting many cells far off the diagonal and is particularly beneficial when sequences are similar, i.e., $s$ is small. Since the WFA algorithm [71] was published, there have been several follow-up studies to further reduce the time and memory [24, 70]. Notably, the BiWFA algorithm [70] reduces the space complexity to $O(s)$, which improves on the previously known lower memory bound $O(n)$ [76]. A current result shows that we can exploit the massive parallel capabilities of modern GPU devices to accelerate WFA [2]. Currently, different implementations exist that have been tested on long reads [71][4], although no dedicated long-read mapper has integrated them yet.

---

[4] https://github.com/waveygang/wfmash/blob/master/README.md, https://github.com/lh3/miniwfa

### Gap cost models

Classical pairwise alignment methods [36, 77] typically relies on global alignment using algorithms derived from Needleman and Wunsch [78]. These alignment algorithms are based on alignment matrices, which aggregate the base-wise alignment scores from the two prefixes (top left of the matrix) to the two suffixes (bottom right). The optimal alignment may look very different depending on which type of penalties are used for gaps in the alignment. Two natural formulations are either using a constant gap penalty, where the gaps get the same fixed penalty $A$ regardless of gap length, or a linear penalty, where the penalty $B$ is multiplied by the gap length $L$). Neither of these gap cost models captures very well the error profiles of long reads and characteristics of indels. While error rates of long reads are constantly changing, traditionally, the long reads had frequent shorter indels. In addition, longer insertions and deletions exist as natural sequence variation. Due to this, it seems natural to introduce a slightly higher cost at opening a gap but not penalizing successive gaps as much. Therefore, the most popular gap cost penalty in long-read aligners is the gap-affine penalty, which combines fixed and linear costs. The gap affine penalty is defined as $A + B(L - 1)$, where $A$ is typically a much higher number than $B$. This gap cost, to some extent, models the nature of long-read errors and smaller indels and furthermore does not increase runtime over the linear cost model due to Gotoh's formulation [36]. In practice, a 2-piece affine gap cost model is typically used to score shorter and longer indels differently [59] and enables alignments over longer indels. However, a 2-piece gap cost model is not a nuanced representation (model) of gaps of different lengths (e.g., due to indels). There are formulations that better model indel lengths, such as a concave gap cost ($A + B \ln L$). Such a model is used in the long-read aligner NGMLR and is computationally more demanding but improves structural variation detection [89].

### Vectorization for speed-up

The extension alignment step is commonly accelerated through vectorization using single instruction multiple data (SIMD) sets of instructions [19, 28], which increase the computational throughput by passing several matrix cells for the processors to evaluate simultaneously in one instruction. Typical SSE architectures have 128-bit registers, which allow sixteen 8-bit matrix cells (or eight 16-bit cells) to be processed in parallel, and more recent AVX go up to 256-bit registers. Thus, traditional SIMD implementations [28] can achieve eight or sixteen times parallelization for short sequences where scores do not exceed the 8-bit or 16-bit limit, being 127 and 32,767, respectively. Under most gap penalties, longer alignments can exceed this value and would therefore require 32-bit values for matrix cells. However, in [96], the authors introduced a difference-recurrence-based formulation that allows storing only differences between the values of adjacent cells in the matrix, which can be represented in only 8 bits, restoring the usage of 8-bit matrix cells even for larger alignments. To our knowledge, all competitive long-read aligners use extension modules that allow vectorization to speed up the extension process.

### Heuristics for speed-up and quality enhancement

Considering the full matrix is unnecessarily expensive in practice in many cases. Practical alignment implementations relies on banded alignment [68, 80], which, simply put, bounds the alignment matrix in a band of size $\ell$ around the top-left – bottom-right diagonal. There are also exit-early strategies. Inspired from BLAST's X-drop [4, 59] implements a Z-drop procedure. X-drop quits extending the alignment if the maximum score reached at some point when aligning the prefix drops by more than X. Z-drop adds the possibility not to drop the extension during large gaps.

Due to sequencing errors, some spurious anchors main remain in a chain, which can lead to a sub-optimal alignment. At the alignment step, [59] chooses to remove anchors that produce an insertion and a deletion at the same time (>10bp) or that lead to a long gap at the extremity of a chain. Another solution [15] involves to re-compute a chain with novel anchors computed on a window that comprises the alignment.

### Future directions

The seed-chain-extend methodology has remained a popular approach in long read mapping since the start of long-read sequencing technologies. There has also been recent advancement on the theoretical side. In [92], the authors show that long read mapping using the seed-chain-extend method is both fast and accurate with some guarantees on the average-case time complexity. We therefore believe this methodology will continue to be a popular approach in the domain of long read mapping.

We have also seen several recent advances in e.g, the seeding and extension steps. Novel seeding techniques such as syncmers [22, 23], *k*-min-mers [25], strobemers [31, 84] have already led to the emergence of practical solutions for robustness to errors and mutations. Additionally, the usage of *diagonal-transition algorithms* for the gap-affine model, which was initially define for edit distance [42, 55, 97], has been reactivated with the wavefront alignment algorithm (WFA, including [24, 70, 71]), offering the potential to make the extension step faster.

Nevertheless, despite these recent advancements, the individual steps of seeding, chaining, and extension must be adapted to several upcoming challenges. For example, we are witnessing a drastic change in the typical reference sequence to which reads are mapped. We will witness more complete genomes that include challenging regions such as centromeres (the T2T consortium recently made available a complete human genome [79]), novel applications are being developed that focus on mapping over difficult instances like repetitive loci [12] and complex structural variants [82] and haplotypes [11]. Adapting and tailoring long-read aligners to such applications will significantly improve analysis over the limited possibilities existing with short reads. Moreover, using pangenomes represented as graphs made from a set of reference genomes is becoming more prevalent [37, 41, 63]. As a result, long-read mapping to these structures is a novel and active field for genomic reads but should soon expand to other applications such as transcriptomics [93]. Notably, pangenome graphs vary in definition and structure (overlap graphs, de Bruijn graphs, graphs of minimizers) and therefore expect a diversified algorithmic response to mapping

sequences on these graphs. Novel challenges lie again in indexing and maintaining a good alignment accuracy despite the accumulation of variations as the number of genomes increases.

## Supplementary information

---

**Additional file 1.** Additional figures and details on tools' strategies.

**Additional file 2.** Review history.

---

## Acknowledgements
The authors would like to thank Mikaël Salson and Laurent Noé for proofreading the manuscript and suggesting revisions.

## Peer review information

## Review history
The review history is available as Additional file 2.

## Authors' contributions
All authors contributed to the writing of the manuscript, with KS making a significant impact on the text. TB and CM proposed the figures included in the manuscript. BC played an important role in the chaining section, leading the effort and proposing the tool. CM led the project. The author(s) read and approved the final manuscript.

## Funding
This work was funded by ANR INSSANE ANR21-CE45-0034-02 project. Kristoffer Sahlin was supported by the Swedish Research Council (SRC, Vetenskapsrådet) under Grant No. 2021-04000.

## Declarations

### Competing interests
The authors have no competing interests to declare.

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

## 

