## [**Additional file 2.** Review history. · Genome Biology]

Review History

First round of review

Reviewer 1

Comments to author:

The manuscript covers a detailed review of algorithmic ideas that were specifically developed for long-read mapping. As large number of papers have been published on this topic, a review article summarising them is timely. The authors do a good job in categorising the methods in a hierarchical manner. Graphic illustrations used by authors are good-quality and will be helpful for readers. The list of surveyed long-read mapping papers is thorough and nearly complete. Unfortunately, I think the quality of writing does not meet acceptable standards in many sections. I would suggest such parts of the manuscript should be re-written carefully.

-- It would help to make algorithm descriptions more concise, especially in subsections between 3.1-3.2. Subsections "Fuzzy seeds handling substitutions" or "Maximal Exact Matches" can probably be removed.

- "Sketching": I am not sure how authors have distinguished between sketching and subsampling. I assume any method that uses a subset of input (or compact fingerprints of sequences) is a type of a sketching method. Please fix the terminology.

-- For fuzzy seeds, the benefit of linking is unclear. What is the difference between linking seeds, and the chaining concept? Which one is better?

-- Section 3.3.2 "Let each anchor be mapped to 1..n integers". How is the mapping done? An anchor was previously defined as pairs of intervals on the query and the reference sequence.

The connection of chaining to the longest increasing subsequence is unclear. A formal definition of the chaining may help. Also, how are "n" and "c" defined?

Line 422: "non-overlapping anchors" The meaning of overlap between anchors is not defined.

-- Line 420: "If inexact matches in seeds are possible, find a series of anchors ensuring a minimal Levenshtein distance between the query and the reference." Again, the meaning of this sentence is unclear

-- Line 527: "this time complexity can be improved in $O(c)$ "; this is unclear. As per definition 3, there is at least a sorting step needed. How is linear-time achieved?

In Definition 3, c is the size of the chain and k is the size of array A. Shouldn't the runtime depend on k as well?

-- Line 495 "With $f(i), f(j)$ scores for anchors i and j," These notations are not defined properly. "The score $f(i+1)$ represents the cost of appending an anchor a_{i+1} to a_i to the chain. Meaning of this sentence is ambiguous.

-- Line 535 "Primary chains are those with the highest scores which do not overlap with another ranked chain for the most of their length." What does overlap mean here?

-- A couple of existing ideas in the context of long-read mapping are missed in the review:

(a) 2-piece affine gap cost model to improve the extend stage (Li 2018)

(b) Difference recurrence relations, and adaptive banding concept for pairwise alignment (Suzuki and Kasahara 2018)

(c) Learned index data structure to accelerate minimizer lookup for the seeding stage (Kalikar et al. 2022)

Minor:

- The web tool is missing coordinates on x and y axis

- Line 68: q' and t' should be characters and not strings if they belong to the alphabet set. There is a typo here.

- Figure 1 (seeding/chaining) have different anchor positions; it would be better to make it consistent

- can find better alternative to the word "destroyed" throughout the manuscript, e.g., corrupted , altered?

- Figure 6 appears much later from the section where it is referred

- Figure 4; the purpose of green line is unclear. Consider removing if it helps simplifying this figure.

- Fix grammar throughout the paper. A few typos are listed below:

-- Definition 3; it is not clear what is "above and to the right" because each anchor is a line segment and not a point

-- Line 143: Is chain a subsequence or a subset of anchor set A?

-- Line 149 : "the continuous work effort put"

-- Line 168 : paragraph ends with word "and"

-- Line 209 : "sketching Sketching"

-- Line 214 : "to perform genome-length sequences alignment-free mapping"

-- Line 308: "Strobemers have shown effective at"

-- Line 310: "but they come at an increased computational cost to joining neighboring minimizers." *due to*?

-- Line 342: "The procedure finds smallest subsequences" subsequences or substrings?

-- Line 344: "In practice, the more the repeats are divergent, the shortest the MCASs"

-- Line 412: "Moreover, over parameters have to be taken into account"

-- Line 460: "one wants to obtain the closest base-wise anchor chain"

-- Line 476 "It is globally the same spirit as LIS"

-- Line 496 "he distance between the two anchors must be $> G$ ", should this be $<?$

Reviewer 2

Comments to author:

This review covers important developments in long read alignments from the BLASR to popular modern tools such as minimap2 and winnowmap2. It focuses on approaches for aligning long sequences to references, although some salient details from read-overlap algorithms from assemblers are also discussed. The authors go into extensive detail about seeding strategies from MEMs to minimizer approaches and cover current developments such as stroblers, syncmers, and MCASs. Modern chaining approaches are presented, and finally a short section about extending alignments.

Overall, I found the material sufficiently accessible and fairly well organized, although I think the article could benefit from a little streamlining (some minor comments below). To my knowledge, there is not a comparable review covering these topics, and quite a bit of recent progress is captured in this manuscript.

Major comments:

I am not sure why ngmlr [69] was relegated to a short statement about structural variant identification (line 153-155). It has been largely replaced, especially since Sniffles2 was developed using minimap2 alignments (<https://www.biorxiv.org/content/10.1101/2022.04.04.487055v1.full.pdf>, page 22). The seeding and chaining strategy is different enough, however, that it deserves mention, even if briefly. Sniffles/ngmlr does have almost 900 citations. Similarly, I think LAMSA deserves a little more credit. Yaha (PMID 22829624) actually pre-dates BLASR by publication date, and while it never achieved so much prominence, it is an early attempt at long-read alignment and should probably be included, at least briefly, where relevant.

The review also focuses on seeding through variants and repetitive regions, which is consistent with developments in long-read alignment strategies. However, I think some other important topics are given less attention. The most prominent is aligning through structural variants and cost functions (related to comments above on NGMLR). The review could benefit from a clearer

discussion on gap models (affine, 2-piece affine in minimap2, and concave models in minimap2, ngmlr, and LRA) and their impacts on efficiency.

Recent work on improving alignments through large repeats, specifically TandemAligner (10.1101/2022.09.15.507041, and earlier TandemMapper work), and how these approaches differ from standard tools would be beneficial to some readers, although I am a little dubious about the claims made by these papers. A small section about more specialized methods like these might be useful especially since centromeres are no longer effectively decoys, but instead are becoming important substrates for variant detection.

The discussion is lacking some content, for example, a broad overview of where long-read alignments are going. I think two clear trends are emerging, namely 1) better seed and alignments over repetitive loci, and 2) ability to cover larger and more complex structural variants, which are two key areas long reads clearly dominate over short reads with optimal methods for traversing them. This is an under-explored space, especially with longer more accurate reads, better assemblies, and references that include centromeres and acrocentric arms (T2T). Long-read alignments to pangenomes is also an active area of development with potentially exciting applications. Please use your own judgment on how to formulate a discussion, but do bring these ideas together with a perspective on future developments.

Minor Comments:

* 64-76: I am not sure that rigorous definitions are needed since the main paper sparsely relies on them. For example, "n" and "l" could be defined more succinctly to support the asymptotic alignment behavior on line 83. If space is needed, consider moving rigorous definitions to methods.

* 112-113: Because sketching is so ubiquitous in modern aligners, I think starting with a clear definition for what it is and a brief motivation for what it is trying to accomplish will help many readers who are vaguely familiar with it.

* 123-124: The statement about MHAP overcoming a BLASR limitation appears correct, but oddly worded since MHAP was designed for read overlaps, not references. Maybe "...which would index full sequences."? Check, but it may not need revising.

* 126-127, 215-216: Another important limitation worth highlighting here is that a fixed-length minimizer doesn't scale with sequence size or length differences between sequences.

* 141: "be an list" should be "be a list"?

* 143: "sorted by such that", remove "by".

* 168: Sentence truncated.

- * 183: I may be a completely naive on this point, but I thought a single substitution would destroy k k-mers (one for each position the substitution appears in the k-mer). where does 2k-1 come from? Am I missing it in Belbasi 2022 [6] (it's not a paper I am familiar with).
- * 215: MHAP was also used for read-to-read overlaps in PBcR-MHAP, which is also in [7]. Might be appropriate to cite again here (i.e. "[7,70]").
- * 238: I think this would be a good place for Heng Li's response to winnowmap (citation [45], section 2.1) before syncmers. It's not mentioned until line 392 (3.2.3).
- * 256-267: Since spaced seeds are not used by long-read aligners, this section could be condensed if needed to make space elsewhere.
- * 358: Homopolymer compression is also heavily used in HiFi, this sentence makes it sound like it's more of an ONT problem. minimap2 explicitly supports this for "PacBio"; 'minimap2 -h' outputs "-H use homopolymer-compressed k-mer (preferrable for PacBio)" (v2.24).
- * 364: "Homopolymer compression is ubiquitous in long-read mappers.". I think homopolymer compression is more common in assembly overlap stages than read or assembly alignments, but some aligners do support it (i.e. minimap2). Please at least cite appropriate tools to support the statement.
- * 427-428 (figure legend): A4 does not have a green line. Maybe "The line passing by the longest chain is shown in green if a shorter chain is selected".
- * 412: "Moreover, over parameters", do you mean "other" instead of "over"?
- * 428-429: Suggest changing "1-" and "2-" for clarity, looks like a hyphenated term. Alternatives could be to "1)" and "2)" or "1 - " and "2 - " (using endash).

Reviewer 3

Comments to author:

In recent years, the field of mapping long reads, which are a type of data generated by third-generation sequencing technology, to a reference genome has seen significant progress. This has been driven by the unique characteristics of long reads compared to shorter reads, which required the development of new mapping methods using a "seed-and-chain" framework rather than the traditional "seed-and-extend" framework. There are now many tools available for mapping long reads, and the field has made rapid advancements with regular updates to data and methods. The authors of "A survey of mapping algorithms in the long-reads era" have written a survey article that provides an overview of existing methods for mapping long reads, focusing on making the methods easy to understand. They also note that the implementation of these methods plays a significant role in the mapping process and have included an original visualization tool to help users understand the parameter settings used in the chaining part of these methods. The paper is well-written, and the authors clearly and concisely introduce the "seed-and-extend" and "seed-and-chain" frameworks and other mapping terms and algorithms, including illustrations to aid in understanding.

Dear editors and reviewers,

Thank you for your careful consideration of our manuscript. We appreciate the time and effort you and the other reviewers have taken to provide us with your valuable feedback. We are pleased to hear that the reports are broadly favorable and that the editor is interested in publishing the manuscript. We understand that the issues raised must be addressed in full, and we committed to addressing each of the points raised in a revised manuscript. In the following you will find a point by point response to the reviews.

We look forward to your feedback on the revised version.

Sincerely,
Camille Marchet on behalf of all authors.

Reviewer 1

-- It would help to make algorithm descriptions more concise, especially in subsections between 3.1-3.2. Subsections "Fuzzy seeds handling substitutions" or "Maximal Exact Matches" can probably be removed.

Response: We followed the reviewers suggestion and streamlined and shortened the text, as well as removing the sections "Fuzzy seeds handling substitutions" and "Maximal exact matches". While the revised text in section 3.1 and 3.2 is only half a page shorter, we have added a discussion of two recent approaches TensorSketch and k-min-mers that increased the length of the sections.

-- "Sketching": I am not sure how authors have distinguished between sketching and subsampling. I assume any method that uses a subset of input (or compact fingerprints of sequences) is a type of a sketching method. Please fix the terminology.

Response: We thank the reviewer for the remark. We decided to use only the term sketching in our manuscript and have avoided to use the term 'subsampling' as a collective term to classify a number of seeds. We have revised and streamlined section 3.1.2 in regards to this change.

-- For fuzzy seeds, the benefit of linking is unclear. What is the difference between linking seeds, and the chaining concept? Which one is better?

Response: With linking, we mean the process to hashing several seeds into one hash value, before it is inserted in the index (i.e. we 'join' several k-mers into a seed). Such linking creates a fuzzy seed with the advantage of being tolerant to indels and can also span large regions, and thus, offer increased uniqueness over single shorter k-mers. In addition, such a seed requires only one query to the index. Linking of seeds happens in the index creation and during the seeding step in the algorithm.

Chaining happens after anchors (seed matches) have been found. Therefore, is it possible to perform chaining of linked seeds.

We aimed to clarify this by adding the sentence "To remove the overhead of post-processing of nearby seeds, one can instead link several k-mers into a seed and represent it as a single hash value before storing it in the index."

-- Section 3.3.2 "Let each anchor be mapped to 1..n integers". How is the mapping done? An anchor was previously defined as pairs of intervals on the query and the reference sequence.

-- The connection of chaining to the longest increasing subsequence is unclear. A formal definition of the chaining may help. Also, how are "n" and "c" defined?

Response: The sentences that the reviewer is referring to have been modified to clarify the idea of usage of LIS.

-- Line 422: "non-overlapping anchors" The meaning of overlap between anchors is not defined.

Response: We specified what overlap means directly after the anchor definition, "two anchors are said to overlap if one anchor starts or ends within the coordinate interval defined by the other anchor".

-- Line 420: "If inexact matches in seeds are possible, find a series of anchors ensuring a minimal Levenshtein distance between the query and the reference." Again, the meaning of this sentence is unclear.

Response: Levenshtein distances are computed for each seed using corresponding intervals on the target and query. We clarified the sentence.

-- Line 527: "this time complexity can be improved in $O(c)$ "; this is unclear. As per definition 3, there is at least a sorting step needed. How is linear-time achieved?

Response: The improvement of time complexity comes from the usage of dynamic data structure of range maximum queue instead of static binary tree. The sentence has been modified to clarify the idea of the improvement.

-- In Definition 3, c is the size of the chain and k is the size of array A . Shouldn't the runtime depend on k as well?

Response: In practice, works as minimap have shown that chain size could be capped without major degradation of the accuracy of the alignment. Therefore we can consider that c is the main parameter at stake for the runtime, practically independent from k .

-- Line 495 "With $f(i), f(j)$ scores for anchors i and j ," These notations are not defined properly. "The score $f(i+1)$ represents the cost of appending an anchor a_{i+1} to a_i to the chain. Meaning of this sentence is ambiguous.

Response: we corrected these sentences.

-- Line 535 "Primary chains are those with the highest scores which do not overlap with another ranked chain for the most of their length." What does overlap mean here?

Response: We clarified the sentence. An overlap is defined with respect to the chains' coordinates on the reference.

- A couple of existing ideas in the context of long-read mapping are missed in the review:
 - (a) 2-piece affine gap cost model to improve the extend stage (Li 2018)
 - (b) Difference recurrence relations, and adaptive banding concept for pairwise alignment (Suzuki and Kasahara 2018)
 - (c) Learned index data structure to accelerate minimizer lookup for the seeding stage (Kalikar et al. 2022)

Response: We added a much more detailed text about extension alignment (section 4) including citations a and b and several other studies on this topic. As for citation c, we added it to section 3.2.2.

Minor:

- The web tool is missing coordinates on x and y axis
- Line 68: q' and t' should be characters and not strings if they belong to the alphabet set. There is a typo here.
- Figure 1 (seeding/chaining) have different anchor positions; it would be better to make it consistent
- can find better alternative to the word "destroyed" throughout the manuscript, e.g., corrupted , altered?
- Figure 6 appears much later from the section where it is referred
- Figure 4; the purpose of green line is unclear. Consider removing if it helps simplifying this figure.
- Fix grammar throughout the paper. A few typos are listed below:
 - Definition 3; it is not clear what is "above and to the right" because each anchor is a line segment and not a point
 - Line 143: Is chain a subsequence or a subset of anchor set A?
 - Line 149 : "the continuous work effort put"
 - Line 168 : paragraph ends with word "and"
 - Line 209 : "sketching Sketching"
 - Line 214 : "to perform genome-length sequences alignment-free mapping"
 - Line 308: "Strobemers have shown effective at"
 - Line 310: "but they come at an increased computational cost to joining neighboring minimizers." *due to*?
 - Line 342: "The procedure finds smallest subsequences" subsequences or substrings?
 - Line 344: "In practice, the more the repeats are divergent, the shortest the MCASs"
 - Line 412: "Moreover, over parameters have to be taken into account"
 - Line 460: "one wants to obtain the closest base-wise anchor chain"
 - Line 476 "It is globally the same spirit as LIS"
 - Line 496 "the distance between the two anchors must be $> G$ ", should this be $<?$

Response: We addressed all the minor comments and fixed several typos throughout the manuscript.

Reviewer 2

- I am not sure why ngmlr [69] was relegated to a short statement about structural variant identification (line 153-155). It has been largely replaced, especially since Sniffles2 was developed using minimap2 alignments (<https://www.biorxiv.org/content/10.1101/2022.04.04.487055v1.full.pdf>, page 22). The seeding and chaining strategy is different enough, however, that it deserves mention, even if

briefly. Sniffles/ngmlr does have almost 900 citations. Similarly, I think LAMSA deserves a little more credit. Yaha (PMID 22829624) actually pre-dates BLASR by publication date, and while it never achieved so much prominence, it is an early attempt at long-read alignment and should probably be included, at least briefly, where relevant.

Response: We thank the reviewer for these remarks.

1. *We have added the citation to YAHA as one of the earliest long-read aligners.*
2. *We now also mention LAMSA and NGMLR in section 4 (extension alignment) and we have given more credit to the gap model used in NGMLR.*

-- The review also focuses on seeding through variants and repetitive regions, which is consistent with developments in long-read alignment strategies. However, I think some other important topics are given less attention. The most prominent is aligning through structural variants and cost functions (related to comments above on NGMLR). The review could benefit from a clearer discussion on gap models (affine, 2-piece affine in minimap2, and concave models in minimap2, ngmlr, and LRA) and their impacts on efficiency.

Response: We thank the reviewer for these remarks. We have substantially expanded the discussion in our extension level alignment section (section 4). We discuss recent approaches, gap cost models, vectorization and heuristics.

-- Recent work on improving alignments through large repeats, specifically TandemAligner (10.1101/2022.09.15.507041, and earlier TandemMapper work), and how these approaches differ from standard tools would be beneficial to some readers, although I am a little dubious about the claims made by these papers. A small section about more specialized methods like these might be useful especially since centromeres are no longer effectively decoys, but instead are becoming important substrates for variant detection.

Response: We thank the reviewer for this suggestion, though we would prefer keeping our text as is. We specify that we cover only genomic mapping in the introduction, as we believe the paper is dense enough. Specialized methods mentioned by the reviewer are mentioned at the end of section 2.2 ("While this survey covers the genomic mapping aspects, other important contributions...").

-- The discussion is lacking some content, for example, a broad overview of where long-read alignments are going. I think two clear trends are emerging, namely 1) better seed and alignments over repetitive loci, and 2) ability to cover larger and more complex structural variants, which are two key areas long reads clearly dominate over short reads with optimal methods for traversing them. This is an under-explored space, especially with longer more accurate reads, better assemblies, and references that include centromeres and acrocentric arms (T2T). Long-read alignments to pangenomes is also an active area of development with potentially exciting applications. Please use your own judgment on how to formulate a discussion, but do bring these ideas together with a perspective on future developments.

Response: We are grateful for these suggestions. We improved the discussion section by adding to algorithmic challenges the insights on future applications.

-- 215: MHAP was also used for read-to-read overlaps in PBcR-MHAP, which is also in [7]. Might be appropriate to cite again here (i.e. "[7,70]").

Response: Thank you, we applied this correction.

-- 238: I think this would be a good place for Heng Li's response to winnowmap (citation [45], section 2.1) before syncmers. It's not mentioned until line 392 (3.2.3).

Response: Yes, we modified the text accordingly.

-- 256-267: Since spaced seeds are not used by long-read aligners, this section could be condensed if needed to make space elsewhere.

Response: We thank the reviewer for this remark. First, we want to note that spaced seeds has been used in the long read aligner GraphMap (Sovic 2016). We have addressed this more clearly in the subsection on fuzzy seeds, and we have updated Figure 2 to reflect this. We have also, per reviewers request, condensed the fuzzy seeds section and merged it with "fuzzy seeds over indels" into one slightly shorter subsection.

-- 358: Homopolymer compression is also heavily used in HiFi, this sentence makes it sound like it's more of an ONT problem. minimap2 explicitly supports this for "PacBio"; 'minimap2 -h' outputs "-H use homopolymer-compressed k-mer (preferrable for PacBio)" (v2.24).

Response: We amended the text to address that problem.

-- 364: "Homopolymer compression is ubiquitous in long-read mappers.". I think homopolymer compression is more common in assembly overlap stages than read or assembly alignments, but some aligners do support it (i.e. minimap2). Please at least cite appropriate tools to support the statement.

Response: We nuanced our statement in line with this remark, and added a citation.

-- 427-428 (figure legend): A4 does not have a green line. Maybe "The line passing by the longest chain is shown in green if a shorter chain is selected".

-- 412: "Moreover, over parameters", do you mean "other" instead of "over"?

-- 428-429: Suggest changing "1-" and "2-" for clarity, looks like a hyphenated term. Alternatives could be to "1)" and "2)" or "1 - " and "2 - " (using endash).

Response: we applied the corrections suggested for minor corrections.

Reviewer 3

Response: Thank you reviewer 3 for your positive assessment.

Second round of review

Reviewer 1

All my concerns have been addressed. Thanks to authors for doing a meticulous job in this revision.

Reviewer 2

This review is more complete with the additional work authors have put in including more recent advances.

The manuscript still needs some careful proofreading before it should be published, particularly in the revised text, but not exclusively.

A few examples are:

4-13) Odd wording: "In particular, the continuous work effort provided in minimap2...". Did you mean "continuous work provided" or "continuous effort provided"?

5-21) "and methods that does not." "does" should be "do".

5-28) "methods typically stores". Should be "store"

9-4) Missing space "compressionappears"

9-4) "long-read mappers" should be singular ("mapper").